# Exploring the Therapeutic Potential of Ectoine in Duchenne Muscular Dystrophy: Comparison with Taurine, a Supplement with Known Beneficial Effects in the *mdx* Mouse

**DOI:** 10.3390/ijms23179567

**Published:** 2022-08-24

**Authors:** Caroline Merckx, Jana Zschüntzsch, Stefanie Meyer, Robrecht Raedt, Hanne Verschuere, Jens Schmidt, Boel De Paepe, Jan L. De Bleecker

**Affiliations:** 1Department of Neurology, Ghent University and Ghent University Hospital, 9000 Ghent, Belgium; 2Department of Neurology, University Medical Center Göttingen, 37075 Göttingen, Germany; 34BRAIN, Department Head and Skin, Ghent University, 9000 Ghent, Belgium; 4Unit of Molecular Signaling and Cell Death, Center for Inflammation Research, Flemish Institute for Biotechnology, 9052 Ghent, Belgium; 5Department of Neurology and Pain Treatment, Immanuel Klinik Rüdersdorf, University Hospital of the Brandenburg Medical School Theodor Fontane, 15562 Rüdersdorf bei Berlin, Germany; 6Faculty of Health Sciences Brandenburg, Brandenburg Medical School Theodor Fontane, 15562 Rüdersdorf bei Berlin, Germany

**Keywords:** Duchenne, *mdx*, osmolyte, taurine, ectoine, histopathology

## Abstract

Duchenne Muscular Dystrophy (DMD) is a debilitating muscle disorder that condemns patients to year-long dependency on glucocorticoids. Chronic glucocorticoid use elicits many unfavourable side-effects without offering satisfying clinical improvement, thus, the search for alternative treatments to alleviate muscle inflammation persists. Taurine, an osmolyte with anti-inflammatory effects, mitigated pathological features in the *mdx* mouse model for DMD but interfered with murine development. In this study, ectoine is evaluated as an alternative for taurine in vitro in CCL-136 cells and in vivo in the *mdx* mouse. Pre-treating CCL-136 cells with 0.1 mM taurine and 0.1 mM ectoine prior to exposure with 300 U/mL IFN-γ and 20 ng/mL IL-1β partially attenuated cell death, whilst 100 mM taurine reduced MHC-I protein levels. In vivo, histopathological features of the tibialis anterior in *mdx* mice were mitigated by ectoine, but not by taurine. Osmolyte treatment significantly reduced mRNA levels of inflammatory disease biomarkers, respectively, CCL2 and SPP1 in ectoine-treated *mdx* mice, and CCL2, HSPA1A, TNF-α and IL-1β in taurine-treated *mdx* mice. Functional performance was not improved by osmolyte treatment. Furthermore, ectoine-treated *mdx* mice exhibited reduced body weight. Our results confirmed beneficial effects of taurine in *mdx* mice and, for the first time, demonstrated similar and differential effects of ectoine.

## 1. Introduction

The *DMD* gene, with over two million base pairs, is the largest of the human genome. Genetic mutations resulting in the ablation of dystrophin protein give rise to Duchenne Muscular Dystrophy (DMD), a rare muscle wasting disorder with an estimated global prevalence of 4.8 per 100,000 [1,2]. Due to X-linked recessive inheritance, mostly boys are affected by DMD. Chronic inflammation, which is characterized by increased expression of cytokines and infiltrating macrophages in dystrophin deficient muscle, is an important aspect of DMD pathology [3,4]. Besides inflammation, dystrophic muscle typically shows signs of myofiber necrosis, fibrosis and fatty replacement. The progressive nature of DMD causes patients to lose ambulation in their early teens and eventually results in premature death due to cardiac complications or respiratory failure. Fortunately, the use of glucocorticoids and respiratory assistance are able to slow down disease progression and have increased life expectancy of DMD patients from 19–25 years (status before 1970) to approximately 30–40 years nowadays [4,5,6,7]. Although the precise mechanism of action is still elusive, glucocorticoids exert strong anti-inflammatory actions, which is believed to underly the beneficial effect of treatment. However, chronic use of glucocorticoids is associated with important side effects, including weight gain, stunted growth, impaired bone health, and behavioural problems [8].

Taurine or 2-aminoethanesulfonic acid is a natural sulphur-containing amino acid that is either taken up from the diet or synthetized in the body from cysteine or methionine [9]. Besides its function in osmotic homeostasis, the osmolyte taurine participates in protein stabilization and muscle performance [10,11]. Taurine exerts anti-oxidative and anti-inflammatory actions, which is particularly of interest in conditions with an inflammatory pathogenic component, such as DMD. In the *mdx* mouse model for DMD, taurine supplementation alleviated muscle damage and mitigated expression of inflammatory and oxidative stress markers [12,13,14]. Although taurine is considered safe [15], taurine supplementation was associated with a decline in body weight and growth retardation in *mdx* mice [16,17]. Furthermore, taurine treatment was not able to increase taurine levels in *mdx* skeletal muscle tissues. We hypothesized that this might be due to a restricted uptake of taurine by the muscle caused by downregulation of TauT in *mdx* mice, as taurine is a feedback regulator of the taurine transporter (TauT) [18,19].

To overcome these difficulties, we selected ectoine (1,4,5,6-tetrahydro-2-methyl-4-pyrimidine carboxylic acid) as an alternative compound with similar features as taurine. Ectoine is an osmolyte synthetized by bacteria that acts as a protein stabilizer and possesses anti-inflammatory and anti-oxidative activities [20,21,22]. The effect of ectoine has already been assessed in a wide variety of in vitro and in vivo inflammatory disease models with potential applications in Alzheimer’s disease, chronic obstructive pulmonary disease and inflammatory bowel disease [20,23,24]. Moreover, a high tolerability and benign safety profile have been attributed to ectoine [25,26,27,28]. To our knowledge, the compound has not been tested in DMD before.

In this study, the effect of osmolytes on cell viability and inflammation was examined in vitro by exposing cultured human rhabdomyosarcoma cells (CCL-136) to pro-inflammatory cytokines. Secondly, we investigated whether ectoine could serve as an alternative for taurine in the treatment of DMD, a severe muscle disorder associated with chronic inflammation, in vivo, in the *mdx* mouse model. We are the first to evaluate ectoine as a potential treatment for DMD.

## 2. Results

### 2.1. Low Dose Treatment of Taurine and Ectoine Protects against Cell Death in CCL-136 Cells Exposed to Cytokines

The effect of osmolyte pre-treatment on cell death was investigated in CCL-136 cells that were exposed to pro-inflammatory cytokines, e.g., interleukin 1β (IL-1β) and interferon γ (IFN-γ), for 24 h. Cell death, quantified as pixel intensity of propidium iodide (PI)-stained cells, was significantly reduced in the presence of 0.1 mM ectoine (*p* = 0.042) and 0.1 mM taurine (*p* < 0.001), but not in the higher supplement concentrations (Figure 1). Pre-treatment with 100 mM ectoine resulted in an increased pixel intensity of CCL-136 cells that were not exposed to cytokines (*p* < 0.001) (Appendix A).

### 2.2. Osmolyte Treatment Does Not Diminish Gene Expression of Inflammatory Disease Biomarkers in CCL-136 Cells

Gene expression of inflammatory disease biomarkers was studied in vitro using qPCR. CCL-136 cells exposed to inflammatory cytokines (IL-1β + IFN-γ) for 24 h showed a significant upregulation of IL-1β, tumor necrosis factor alpha (TNF-α), interleukin 6 (IL-6), C-C motif chemokine ligand 2 (CCL2) and major histocompatibility complex class 1 (MHC-I) mRNA levels after normalization to the housekeeping gene glyceraldehyde-3-phosphate-dehydrogenase (GAPDH), whereas mRNA expression of Heat Shock Protein Family A Member 1A (HSPA1A) and secreted phosphoprotein 1 (SPP1) was not significantly altered by cytokine exposure (Appendix A). Pre-incubation with taurine or ectoine for 24 h, applied in low (0.1 mM) concentrations, did not alter mRNA levels of inflammatory markers in cells exposured to cytokines, whereas 100 mM taurine increased IL-1β expression (Figure 2).

### 2.3. 100 mM Taurine Alleviates MHC-I Protein Expression in Cytokine Exposed CCL-136 Cells

We examined the effect of osmolyte treatment on MHC-I protein expression in cultured cells using western blot (Figure 3). MHC-I expression is evaluated relative to the untreated cytokine-exposed condition after normalization against total protein using Stain-Free technology (Appendix A). As expected, cells exposed to proinflammatory cytokines (Il-1β and IFNγ) exhibited a significant upregulation of MHC-I protein levels. Treatment with 0.1 mM ectoine in cytokine-exposed cells further increased MHC-I protein levels compared to the untreated cytokine-exposed conditions (*p* = 0.036). Of note, a similar trend was observed in cytokine-exposed cells that received 0.1 mM taurine but lacked significance. Pre-treatment with 100 mM taurine significantly reduced MHC-I protein expression in vitro (*p* = 0.026), whereas treatment with 100 mM ectoine shows a similar trend as 100 mM taurine, but was not significant.

### 2.4. Ectoine-Treated Mice Exhibit Reduced Body Weight

Ectoine-treated mice had lower body weight compared to sham-treated *mdx* mice, irrespective of dosage and method of administration (Table 1). Though the reduction in body weight for *mdx* mice treated with ectoine in drinking water or intraperitoneal injection was small, 12% and 7%, respectively, these differences were statistically significant (*p* = 0.001, and, respectively, *p* = 0.006). Overall, *mdx* mice had a shorter posture than control mice (*p* = 0.018), but osmolyte treatment did not affect body length. In *mdx* blood sera, collected prior to sacrifice, CK levels were unaltered.

### 2.5. Ectoine Treatment Attenuates Histopathological Features in the mdx Mouse

Histopathological characterization was carried out as described previously [11]. The amount of healthy fibers, regenerating fibers (centronucleated fibers) and necrotic fibers (macrophage invaded fibers and fibers with loss of structure and/or pale cytoplasm,) was manually counted in three whole muscle haematoxylin-eosin (H&E) stained sections of the tibialis anterior per mouse (Figure 4). No significant effect of ectoine treatment was observed on the amount of necrotic fibers, yet the percentage of healthy fibers was significantly increased upon ectoine treatment through oral (*p* = 0.005) and i.p. (*p* < 0.001) administration. The fraction of regenerating fibers, defined by the presence of central nuclei, was smaller in *mdx* mice that received ectoine treatment (*mdx* ect oral treatment *p* < 0.01 and *mdx* ect i.p. *p* < 0.001). These fibers presumably represent restored previous damage, thus, our results suggest ectoine treatment attenuates dystrophinopathology.

### 2.6. Osmolyte Treatment Did Not Improve Functional Performance of mdx Mice

Functional four limb hanging wire and open field tests were conducted to assess muscle performance and locomotion (Figure 5). The four limb hanging wire was performed twice, at week 4 and week 5. The holding impulse (hanging time × body weight) was significantly longer (*p* < 0.001) in control mice (3891.6 ± 1255.3) compared to sham-treated *mdx* mice (847.4 ± 131.9) at the age of 4 weeks. Oral administration of ectoine in *mdx* mice on the other hand (598.8 ± 195.0) resulted in a significantly shorter holding impulse compared to sham-treated *mdx* mice (*p* = 0.001). The difference between ectoine-treated mice (2859.27 ± 807.7) and sham-treated mice (2214.8 ± 648.8) was abrogated in 5-week-old mice (*p* = 0.277), whereas the difference between sham-treated mice and control mice (6110.7 ± 1884.4) remained significant (*p* < 0.001) at the age of 5 weeks.

In the open field test, the total distance covered was significantly longer in control mice (4047.1 cm ± 251.8, *p* = 0.002) than in sham-treated *mdx* mice (2839.4 cm ± 333.6). Osmolyte treatment did not alter the total running distance in *mdx*.

### 2.7. Osmolyte-Treated Mice Exhibit Reduced Expression of Inflammatory Myopathy Markers

A set of inflammatory muscle disease markers was examined using qPCR in the tibialis anterior of untreated, ectoine-treated and taurine-treated mdx mice (Figure 6) and normalized to the housekeeping gene GAPDH. mRNA expression of CCL2 was significantly lower in ectoine-treated (*p* < 0.001) and taurine-treated (*p* < 0.001) mdx mice compared to untreated mdx mice. Ectoine treatment, but not taurine treatment, downregulated mRNA levels of SPP1 in mdx mice (*p* = 0.016), whereas taurine treatment downregulated expression of HSPA1A (*p* < 0.001), TNF-α (*p* < 0.029) and IL-1β (*p* = 0.004). We evaluated the presence of F4/F80+ cells, indicative for macrophages, and CCL2 by double immunofluorescent staining. We found no co-localization of the pro-inflammatory chemokine CCL2 with the macrophage marker (Appendix A).

### 2.8. Osmolyte Treatment Did Not Alter Protein Levels of TauT or MHC-I

Protein expression of TauT was significantly lower in sham-treated mdx mice compared to controls (*p* = 0.002) and was not affected by osmolyte treatment. MHC-I levels were moderately yet significantly higher in control mice (*p* = 0.008) compared to sham-treated mdx mice, but levels were not affected by osmolyte treatment (Figure 7).

## 3. Discussion

Taurine supplementation has been described to effectively reduce pathological features in *mdx* mice, however, some studies have reported adverse effects on body weight and growth development [16,17,29,30]. The aim of this study was to investigate whether ectoine could represent an alternative for taurine by preclinical investigations in the *mdx* mouse model. Our study unveiled some beneficial effects of ectoine supplementation in terms of histopathological features and inflammation.

Previous to our study, little was known regarding the uptake of ectoine at the skeletal muscle level. Therefore we selected two doses (e.g., ±180 mg/kg and ±1 g/kg ectoine) and we treated animals via two administration routes (through i.p. injection, and supplemented to drinking water, respectively), whereastaurine was supplemented to the drinking water, and the dose (±4.5 g/kg ≈ 2.5%) was chosen within the effective range [16]. In addition, the effects of taurine and ectoine supplementation were evaluated in human rhabdomyosarcoma CCL-136 cells exposed to pro-inflammatory cytokines, as a model for muscle inflammation.

Similar to other reports, creatine kinase levels were unaltered by osmolyte treatment [12,31,32]. However, ectoine significantly improved histopathological features in the *mdx* mouse model. The relative portion of healthy fibers was significantly higher in ectoine-treated *mdx* mice compared to sham-treated mice. Moreover, ectoine treatment resulted in a lower fraction of regenerating muscle fibers, indicative of previous muscle damage. The attenuation of histopathological features is likely caused by ectoine treatment rather than a litter-dependent observation since muscle damage greatly varies between littermates and even within both tibiales anteriores of the same mouse [33]. Whilst others reported improvement of histological features by taurine treatment [12,13], we did not observe a decrease in regenerating fibers by taurine treatment, which is in line with the study of Barker [34]. Thus, ectoine treatment is more effective than taurine at attenuating histopathological features in the mdx mouse model.

Inflammation is an important aspect of DMD pathology. We report that osmolyte treatment reduced mRNA expression of inflammatory markers in *mdx* mice. However, the anti-inflammatory effect was specific for the type of treatment, with broader anti-inflammatory effects exerted by taurine. The pro-inflammatory chemokine *CCL2*, also known as monocyte chemoattractant protein 1 (*MCP-1*), is increased in the muscle of *mdx* mice [11,35,36]. As CCL2 recruits immune cells to sites of muscle injury, it presents a potential biomarker of disease severity in DMD [35]. *CCL2* mRNA levels were remarkably downregulated in ectoine and taurine-treated *mdx* mice, which points to an anti-inflammatory effect for both osmolytes. In additon, ectoine-treated *mdx* mice exhibited significantly lower *SPP1* mRNA levels compared to untreated mice. The latter protein is secreted by macrophages and modifies both the inflammatory and fibrotic process in dystrophin deficiency [36,37,38]. SPP1 is believed to instigate TGF-β levels, however, *TGF-β* mRNA levels were unaltered upon osmolyte treatment. SPP1 inhibition has been shown to improve pathological features in the *mdx* mouse model [38]. *IFN-γ* levels were not affected by ectoine treatment nor taurine treatment. In general, taurine treatment exerted a broader effect on inflammatory disease markers. Besides downregulation of *CCL2* and *HSPA1A*, taurine also significantly reduced expression of pro-inflammatory cytokines *TNF-α* and *IL-1β*. These results point to taurine exhibiting a more potent anti-inflammatory effect than ectoine in *mdx* mice.

As taurine is a known modifier of its transporter, TauT, and taurine supplementation resulted in a significant downregulation of TauT levels in vitro [19], we investigated protein expression of TauT in vivo. Similar to our previous results [11], we observed a significant downregulation of TauT protein levels in *mdx* mice compared to control mice. However, TauT levels did not change upon osmolyte treatment, which is in line with previous findings [17,39]. Expression of MHC-I was significantly higher in control mice compared to *mdx* mice and was not affected by osmolyte treatment. Elevated MHC-I levels in control mice might be explained by basal levels of MyoD+ cells, which express MHC-I in young mice [40,41]. In vitro experiments showed reduced lncRNA MyoD expression in myoblasts of *mdx* mice compared to controls [42] and reduced MyoD expression in *mdx* mice [43], which might point towards decreased levels of MyoD+ cells in *mdx* mice, accompanied by lower MHC-I levels.

In vitro, mRNA expression of inflammatory markers including *MHC-I* was not downregulated by taurine nor ectoine, whereas, at the protein level, MHC-I was significantly lower in cytokine-exposed cells incubated with 100 mM taurine. In contrast to taurine, 0.1 mM ectoine increased MHC-I protein expression. Furthermore, low doses of taurine and ectoine attenuated cell death, while cell death was enhanced in cells treated with a high dose of ectoine in absence of cytokine exposure. Thus, we observed a clear dose-dependent effect of treatment in vitro, with high concentrations of taurine being more effective at reducing MHC-I protein expression and low concentrations of taurine and ectoine attenuating cell death.

We observed a reduction in body weight in osmolyte-treated *mdx*, which would need to be investigated further as this could hinder the use of ectoine as a potential treatment for dystrophin deficiency. Previous reports describe a decreased body weight in animals receiving taurine treatment with doses varying from 20 mg/kg to 16 g/kg [16,30]. Anorexigenic effects have been attributed to taurine, since taurine administration directly in the hypothalamus resulted in a lower food intake [30]. Similarly, food intake was significantly reduced in taurine treated (20 mg/kg) castrated mice but did not affect body weight in this group, whilst body weight was reduced in taurine-treated non-castrated mice in which group the food intake was unaltered [29]. Based on these findings, we hypothesized taurine treatment to affect body weight in *mdx* mice, yet, surprisingly, we found a significant lower body weight in ectoine-treated *mdx* mice only, but not in taurine-treated *mdx* mice. Our results strongly indicate an effect of ectoine on body weight, especially since low body weight was observed in both ectoine-treated groups. To our knowledge, this is the first study describing weight loss in ectoine-treated mice. Previous studies have reported either no effect of ectoine on body weight or weight loss in experimental colitis, counteracted by ectoine treatment [21,44]. The mechanism by which ectoine might exert this effect remains unknown and needs further investigation. A partial explanation could be a litter-dependent effect, since all ectoine-treated pups might have represented descendants of the same mating couple. In addition, litter size, which was the highest in mice receiving ectoine in the drinking water (n = 11), is inversely related to body weight at weaning and thus could also attribute to the low body weight in this group [45]. As treatment was initiated early in life, it was not possible to give littermates different treatments in this set-up, hence, our study could not control such litter-dependent effects. Therefore, the effect of ectoine on body weight should be further investigated in a litter-controlled set-up. We did not observe any effect of treatment on body length, implying that growth rate is unaffected.

Though our study revealed some beneficial effects on muscle pathology, muscle performance and locomotion, evaluated by the four limb hanging wire and open field test, was not improved by osmolyte treatment. At the age of 4 weeks, the holding impulse of *mdx* mice receiving ectoine in the drinking water was diminished compared to those receiving sham treatment, which was then normalized by the age of 5 weeks. Osmolyte treatment did not alter the total running distance of mice in the open field test. Previously, taurine treatment in mdx mice also lacked an effect on running distance during exhaustion tests [31].

In conclusion, this study examined the effects of ectoine and taurine in pro-inflammatory cytokine exposed rhabdomyosarcoma cells in vitro and found a significant dose-dependent effect of osmolyte treatment. A high dose of taurine pre-treatment prevented MHC-I protein upregulation in vitro, whereas low doses of taurine and ectoine attenuated cell death. In addition, we were the first to examine the effects of ectoine on muscle in vivo in the *mdx* mouse model for DMD. Ectoine significantly ameliorated histopathological features in *mdx*, but we did not find any effect of taurine treatment. Both ectoine and taurine treatment effectively reduced gene expression of inflammatory markers in *mdx*, with a broader anti-inflammatory effect established by taurine treatment, yet functional tests could not show improvement of either treatment. A reduction in body weight was observed in ectoine-treated *mdx* mice which could interfere with its potential as a supplementary treatment in dystrophin deficiency. Thus, our results show that ectoine could be of interest as a supportive treatment for DMD, yet additional experiments are required that (i) investigate its effect on body weight and (ii) determine optimal dose and administration routes.

This study has several limitations that should be considered. Firstly, the effect of treatment on muscle inflammation was evaluated in CCL-136 cells exposed to cytokines [46]. In this study, the use of dystrophin-deficient muscle cells exposed to cytokines would have been a more suitable model. However, we experienced problems maintaining primary dystrophin-deficient cells in culture, and these experimental difficulties did not allow us to properly study the effect of treatment in these cells. Secondly, gene expression of inflammatory disease markers was examined in untreated and treated *mdx* mice, but not in control mice. Lastly, this study focussed on short-term effects of treatment, as this was designed as a proof-of-concept for ectoine. Long-term effects of treatment should be evaluated in follow-up studies. In these follow-up studies, evaluation of functional performance in older mice are encouraged, as behavior observed in younger mice can influence results.

## 4. Materials and Methods

### 4.1. Cell Culture

Human rhabdomyosarcoma CCL-136 cells (ATC, Manassas, VA, USA) were grown in Dulbecco’s Modified Eagle Medium (DMEM) (Thermo Fisher Scientific, Waltham, MA, USA) supplemented with 10% fetal calf serum, 1% penicillin/streptomycin (Biochrom, Cambridge, UK) and 1% L-glutamine (200 mM) (Thermo Fisher Scientific, Waltham, MA, USA) and kept under controlled conditions (37 °C, 5% CO_2_) during the entire experiment. Taurine (Merck, Overijse, Belgium) and ectoine (Merck, Overijse, Belgium) were dissolved in PBS and sterilized before cells were incubated for 24 h with either low dose (0.1 mM) or high dose (100 mM) osmolyte solution, or without any addition. Cytokine exposure included 300 U/mL recombinant human Interferon-γ (Invivogen, San Diego, CA, USA) combined with 20 ng/mL recombinant human Il-1β (Invivogen, San Diego, CA, USA), and analysis was performed 24 h after cytokine exposure.

### 4.2. Animals and Drug Regimens

C57BL/10ScSn-Dmdmdx/J (*mdx*) mice and C57BL/10SnJ control mice were bred at the central specific pathogen free animal facility of Ghent University (ECD 17/130). The experimental procedures were performed in accordance with ARRIVE guidelines and approved by the Animal Ethics Committee of Ghent University (ECD 19/110). All animals had access to food and water ad libitum.

A total of six groups were included in the experiment: *mdx* sham group (*n* = 8, 1 litter) that received i.p. saline injection, *mdx* group receiving ectoine in drinking water (*n* = 11; 0.5% ect ≈ 1.08 g/kg, 1 litter), *mdx* group receiving taurine in drinking water (*n* = 11; 2.5% tau ≈ 4.6 g/kg, 2 litters), *mdx* group receiving ectoine i.p. (*n* = 9; ≈ 177 mg/kg ectoine solved in saline daily injected, 1 litter), C57/BL10SnJ control mice (*n* = 7, 1 litter), and *mdx* control group (*n* = 10) that received regular drinking water. Treatment was initiated at postnatal day 7; from then on, drugs were added to the drinking water. Since i.p. injection in *mdx* mice aged 7 days is too risky, this group received ectoine in drinking water (0.075% wt./vol ± 150 mg/kg) until postnatal day 21, and this was then changed to i.p. injection. Weaning was carried out at postnatal day 28. Both male and female mice were included in this study. Throughout the experiment, the weight of mice and the amount of water mice drank were closely monitored, and water intake per weight was approximately the same for *mdx* mice receiving taurine/ectoine compared to control mice that had access to regular drinking water.

At the end of the experiment (day 41 ± 1 day), mice were euthanized by i.p. injection of a mixture of Nimatek (100 mg ketamine/mL, Dechra Pharmaceuticals, Nortwich, UK) and Rompun (2% Xylazine, Bayer, Leverkusen, Germany) in a ratio of 2:1. In the absence of reflexes, blood was collected from the retro-orbital plexus. Blood was allowed to cloth in the dark for over 30 min at room temperature and was centrifuged (15,000 rpm) for 10 min at 4 °C and stored at −70 °C until creatine kinase concentration was determined. Next, cervical dislocation was conducted and the length of mice was measured. Muscles were dissected and frozen using dry ice or using nitrogen-cooled isopentane.

### 4.3. Four Limb Hanging Wire Test

On the day of the four limb hanging wire test, mice were weighed and placed on the grid to get familiarized. The grid was inverted so that mice hung upside down at a height of approximately 50 cm above a cage containing soft bedding material. The duration mice were hanging upside down was timed by the investigator, and a maximum hanging time was set at 600 s. Mice were given 5 attempts with 5–10 min rest, allowing them to recover. The Holding Impulse (body weight × maximum hanging time) was used for analysis. The Four Limb Hanging Wire Test was assessed at day 30 ± 1 day and 38 ± 1 day. Two animals were excluded from analysis as they voluntarily jumped off the grid.

### 4.4. Open Field Test

Mice were individually habituated to the open field cage (60 × 60 cm) for 3 consecutive days for 10 min before data were gathered. At the day of testing (day 39/40), mice were placed into the cage, and their behaviour was recorded for 10 min. Afterwards, videos were analysed using Optimouse Software that was run on Matlab 2017, and the total distance run by mice during this period was used as the outcome measure. One mouse was excluded from analysis as the running distance was <10 cm.

### 4.5. Histology

The tibialis anterior was dissected and immersed in Tissue Tek O.C.T (Sakura, Alphen aan den Rijn, The Netherlands) prior to snap-freezing in isopentane. Next, sections (8 µm) were made using a microtome, stained with H&E and histological analyses was performed by manually counting healthy muscle fibers, regenerating muscle fibers (identified by the presence of one or more centralized nuclei) and necrotic fibers (macrophage invaded fibers, fibers with loss of structure and/or pale cytoplasm) on digitalized sections. In total, three sections of the tibialis anterior (>150 µm apart) were analysed. Per section, an average of 2226 fibers were counted by a blinded investigator, and analysis was carried out as described previously [11].

### 4.6. Immunofluorescence Staining

Sections were briefly exposed to acetone and air-dried prior to blocking with PBS blocking solution (2% bovine serum, 5% donkey serum and 10% heat-inactivated human serum) for 1 h. Sections were incubated with rat F4/F80 antibody (5 µg/mL, ab6640, Abcam, Cambridge, UK) and rabbit CCL2/MCP1 antibody (40 µg/mL, NBP1-07035, Novus Biologicals, Centennial, CO, USA) overnight at 4 °C. Next, sections were rinsed with PBS and incubated for 1 h in PBS containing the AlexaFluor 488 secondary antibody directed against rat and CY3 labelled secondary antibody directed against rabbit. After washing in PBS, sections were mounted with Fluoromount G (Southern Biotech, Birmingham, AL, USA).

### 4.7. Cell Death Assessment

Propidium-iodide staining was carried out according to the manufacturer’s protocol. In short, cells were incubated for 20 min at room temperature with X-vivo medium (Lonza, Basel, Switzerland) containing 0.2% PI (Abcam, Cambridge, UK), and mounted afterwards. Representative grey scale images were taken within 24 h. Background subtraction was carried out prior to calculation of the mean pixel intensity of a subset of representative cells per image and was automatically determined using FIJI (Fiji Is Just ImageJ) software.

### 4.8. RNA Isolation

RNA extraction from the tibialis anterior was carried out as described previously [11]. RNA extraction from the cells was carried out according to the manufacturer’s protocol. In short, cells were rinsed with Dulbecco’s PBS and thereafter incubated with RLT lysis buffer containing 1% β-mercaptoethanol for 30 min at room temperature. Cells were centrifuged at 13,000 RPM and the supernatant was mixed with 70% ethanol and subsequently transferred to the RNEasy spin column. After centrifugation at a speed of 9000 RPM, cells were consecutively rinsed with RW and RPE buffer with intermediate centrifugation steps. The final step included addition of RNAse free water to the spin column and centrifugation at speed 10,000 RPM. The RNA concentration was measured with the Nanodrop 1000 (ThermoFisher) and stored at −70 °C until further processing.

### 4.9. Reverse Transcriptase Quantitative Polymerase Chain Reaction

First, cDNA was prepared according to the manufacturer’s protocol from 200 ng RNA with the use of the PCR Thermal Cycler-Master cycler nexus (Eppendorf, Hamburg, Germany), 500 ng/µL oligodTs, 5× First Strand Buffer, 0.1 M DTT, 10 mM dNTPs and Superscript II Reverse Transcriptase (Invitrogen, Darmstadt, Germany). PCR reactions were run in triplicate on a 7500 Real Time PCR system in the presence of 1 µL cDNA, 10 µL Taqman Gene Expression Mastermix (Applied Biosystems, Foster City, CA, USA), 8 µL RNAse free water and 1 µL of the following primers for CCL-136 cells: GAPDH (hs99999905_m1), IL-1β (hs00174097_m1), TNF-α (hs00174128_m1), IL-6 (hs00174131_m1), CCL2 (hs00234140_m1), SPP1 (hs00959010_m1), HSPA1A (hs00359163_s1), MHC-I, and the following primers were used for mouse tissue: CCL-2 (Mm00441242_m1), SPP1 (Mm00436767_m1), HSPA1A (Mm01159846_s1), TGFβ1(Mm00441724_m1), TNF-α (Mm00443258_m1), IL-1β (Mm00434228_m1), IFN-γ (Mm00801778_m1) and MHC-I. Relative gene expression was normalized to the housekeeping gene GAPDH and expressed as 2^−ΔCT^. Please note, for qPCR experiments, a different *mdx* control group was used that received no sham treatment.

### 4.10. Protein Extraction

Cells were rinsed with Dulbecco’s PBS and exposed to a protease inhibitor (Roche, Indianapolis, IN, USA) supplemented RIPA buffer (50 mm TrisHCl, 150 mM NaCl, 2.5% NP40, 2.5% Na-deoxycholate, 0.1% sodium dodecyl sulphate pH 7.4). Protein extracts were centrifuged for 5 min at a speed of 13,000 rpm at 4 °C. Protein concentrations were measured in triplicate using the Bradford protein assay method carried out in the Infinite M200Pro and analysed with Magellan 7.2 software (Tecan, Mannedorf, Switzerland). The gastrocnemius muscles derived from mice were grounded in the presence of extraction buffer (50 mM TrisHCL, 2 mM EDTA pH 7.4) supplemented with protease inhibitor tablets (Roche, Indianapolis, IN, USA). Next, muscle extracts were centrifuged twice at a speed of 15,000 RPM for 20 min. The protein concentration of the supernatant was measured with Biodrop µLite (Isogen, Utrecht, The Netherlands) and diluted to a protein concentration of 2000 µg/mL. Protein extracts of CCL-136 cells and murine muscle tissue were stored at −70 °C until further processing.

### 4.11. Western Blot

Protein extracts were boiled for 2 min in the presence of loading buffer containing 10% Β-mercaptoethanol and 90% 4x Laemmli sample buffer (Bio-Rad Laboratories, Hercules, CA, USA). Then, 4–20% Mini-Protean TGX Stain-Free gels (Bio-Rad Laboratories, Hercules, CA, USA) were loaded with equal amounts of protein (30 µg or 15 µL) prior to electrophoresis. Next, proteins were transferred onto low fluorescent polyvinylidene membranes (Bio-Rad Laboratories, Hercules, CA, USA) that were pre-treated with ethanol using Trans-Blot Turbo. After protein transfer, Stain-Free blots were imaged by Chemidoc (SF blot pre-defined settings), which were used for protein normalization as described by Bio-Rad. Afterwards, blots were blocked in a tris-buffered saline with 0.1% Tween20 (TBST) blocking solution containing 0.2% non-fat dry milk for 1 h and incubated either for 4 h or overnight in the presence of the following antibodies: anti-GAPDH antibody clone NB615 (1 µg/mL, LS-B9310 Bioconnect, Huissen, The Netherlands) and anti-HLA Class 1 ABC antibody (1 µg/mL ab70328, Abcam, Cambridge, UK) for CCL-136 cells, and for muscle derived from mice: rabbit anti-GADPH antibody-loading control (0.2 µg/mL, ab 9485, Abcam, Cambridge, UK), rabbit anti-slc6a6/TauT antibody (0.125 µg/mL, ab 196821, Abcam, Cambridge, UK), and HLA-ABC rabbit anti-human polyclonal antibody (1 µg/mL, PA598355, Invitrogen). Appropriate secondary antibodies were applied for 1 h. Protein bands were visualized with chemiluminescent Clarity Western ECL substrate (Bio-Rad Laboratories, Hercules, CA, USA) or with chromogenic Western Breeze kit (Invitrogen, Waltham, MA, USA) using the Chemidoc Imaging System (Bio-Rad Laboratories, Hercules, CA, USA). Protein density quantification was carried out with Image-Lab 6.0 software and was normalized to total protein using Stain-Free technology (Bio-Rad Laboratories, Hercules, CA, USA), to correct for loading errors.

### 4.12. Statistics

Data analysis was performed in SPSS Statistics 27 and visualized in GraphPad. Data derived from cell culture experiments were evaluated using a mixed model with the following specifications: the variable ‘treatment’ was defined as a fixed factor and a random subject intercept with the subject ‘plate’ as a nested factor. In vivo experiments were analysed using the univariate ANOVA model with ‘sex’, ‘treatment’ and ‘sex by treatment’ as fixed factors. The overall effect of treatment was reported. Data derived from histopathological analysis (H&E-stained sections) were analysed using the negative binomial log link regression model with ‘sex’, ‘treatment’ and ‘sex x treatment’ as fixed factors, the variable ‘section’ as repeated measure and compound symmetry defined as covariance structure. Data analysis of in vivo experiments included adjustment for differences in sex between groups.

## Figures and Tables

**Figure 1 ijms-23-09567-f001:**
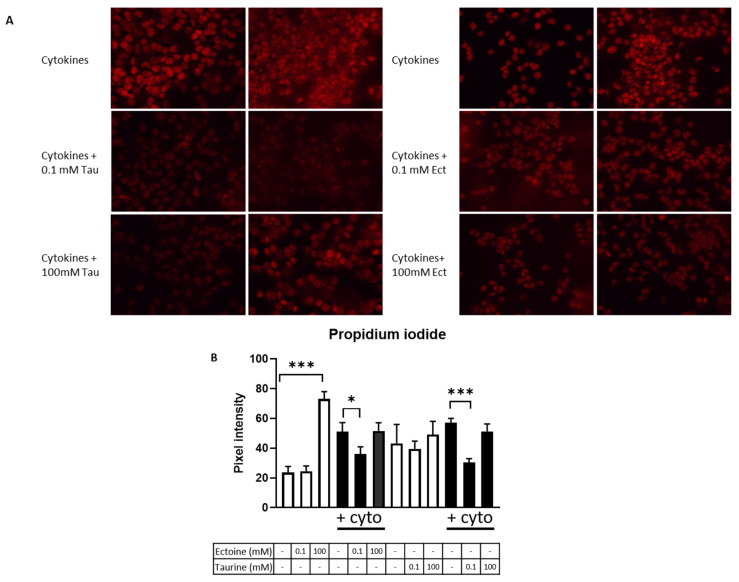
Evaluation of cell death in CCL-136 cells. (**A**) Representative images of PI-stained (red fluorescence) cytokine exposed (IL-1β + IFN-γ) CCL-136 cells that received no pre-treatment, 0.1 mM osmolyte treatment or 100 mM osmolyte treatment. Two representative images are shown per condition. (**B**) Graphical summary of the gray scale analysis performed on PI-stained cells. The abbreviation ‘+cyto’ is used to annotate the conditions in which CCL-136 cells were exposed to cytokines (IL-1β + IFN-γ). Experiments were carried out in triplicate, and data are presented as mean pixel intensity ± SEM. Statistical analysis was carried out in order to detect differences between untreated and treated samples within a specific condition (no cytokine exposure/cytokine exposure), and significant differences are annotated by * (*p* < 0.05), and *** (*p* < 0.001).

**Figure 2 ijms-23-09567-f002:**
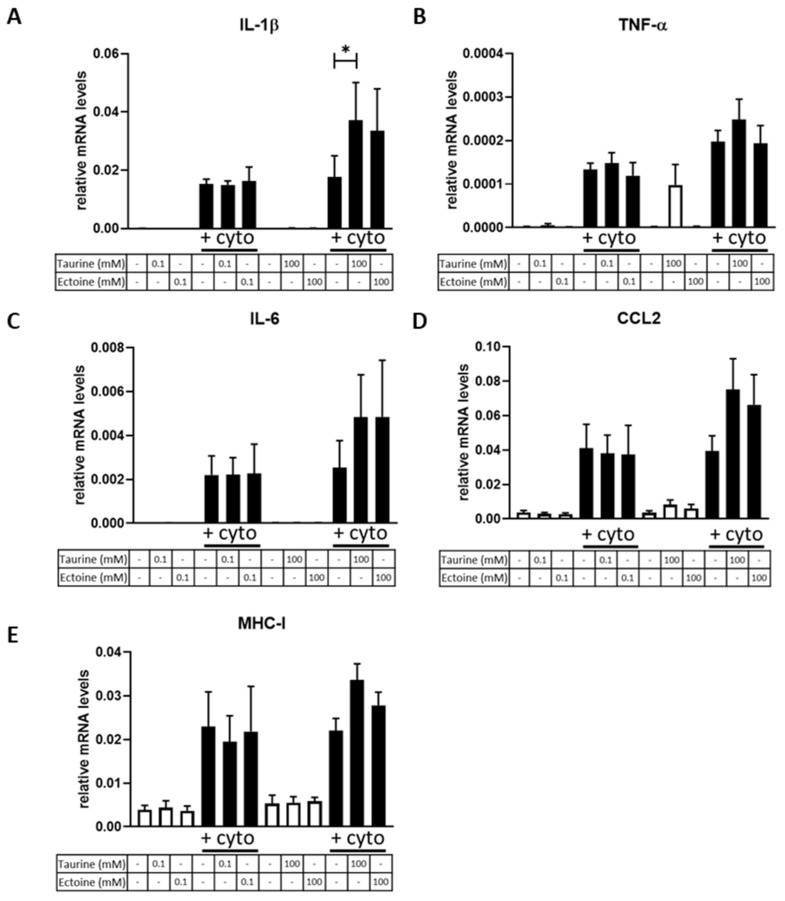
Relative gene expression of inflammatory markers in CCL-136 cells. CCL-136 cells were incubated in medium supplemented with 0.1 mM taurine, 100 mM taurine, 0.1 mM ectoine or 100 mM ectoine, whereas exposure to cytokines (IL-1β + IFN-γ) is annotated by ‘+cyto’. Graphs show mRNA levels of *IL-1β* (**A**), *TNF-α* (**B**), *IL-6* (**C**), *CCL2* (**D**) and *MHC-I* (**E**) after exposure to different treatment conditions. In vitro cell experiments were conducted in triplicate and mean values are depicted in the graphs. Data underwent a log-transformation in order to adhere to normality assumptions. Relative gene expression is normalized to the housekeeping gene GAPDH and expressed as 2^−ΔCT^, and data are presented as 2^−ΔCT^ mean ± standard error of the mean (SEM). Statistical analysis evaluated differences between untreated and treated samples within a specific condition (exposed/not exposed). For clarity, only the significant differences between untreated and treated cytokine exposed samples are shown and were annotated by * (*p* < 0.05).

**Figure 3 ijms-23-09567-f003:**
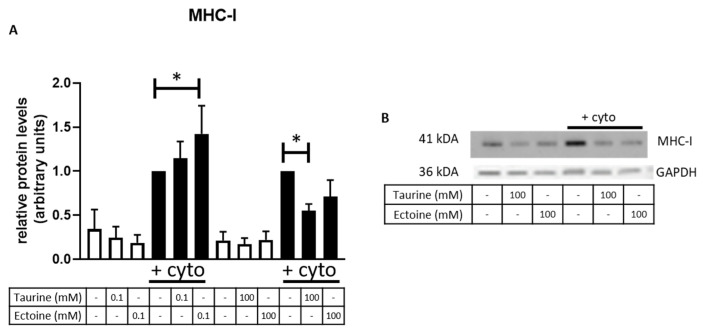
MHC-I protein levels in CCL-136 cells. (**A**) Relative MHC-I protein expression assessed by western blot in CCL-136 cells without treatment and 0.1 mM taurine, 100 mM taurine, 0.1 mM ectoine or 100 mM ectoine treatment prior to cytokine exposure (IL-1β + IFN-γ), annotated by ‘+cyto’, or without exposure to cytokines. In vitro cell experiments were conducted in quadruplicates. Data are presented as mean relative protein expression ± SEM after total protein normalization using the stain-free blot technology. Statistical analysis compared untreated and treated samples within their specific condition (no cytokine exposure/cytokine exposure), and significant differences are annotated by * (*p* < 0.05). (**B**) Representative protein bands of MHC-I and GAPDH are given, as visualized with chemiluminescent and colorimetric detection, respectively, showing equal loading of lanes.

**Figure 4 ijms-23-09567-f004:**
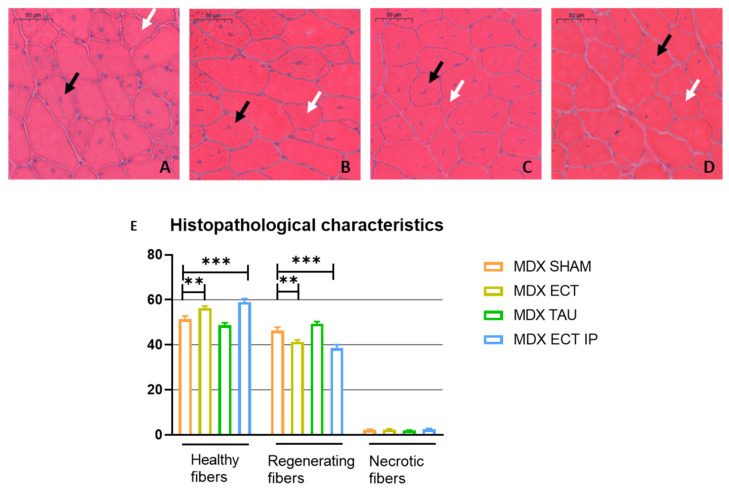
Histopathological features of sham-treated and osmolyte-treated *mdx* mice. H&E-stained section of (**A**) sham-treated *mdx* mice; (**B**) ectoine-treated *mdx* mice through oral administration; (**C**) taurine-treated *mdx* mice through oral administration; (**D**) I.p. ectoine injected *mdx* mice. Healthy (white arrows), regenerating (black arrows) and necrotic fibers (not illustrated) were quantified in three sections per animal (>150 µm apart), yielding analysis of a total of 117 sections. (**E**) Histopathological characteristics of H&E-stained sections of sham-treated (*n* = 8), oral ectoine- treated (*n* = 11) and i.p. ectoine- treated mice (*n* = 9). Data are presented as mean ± SEM, and statistical analysis revealed significant differences between sham-treated *mdx* mice and other groups (Appendix A). Significant differences are annotated by ** (*p* < 0.01) and *** (*p* < 0.001). Scale bar = 50 µm.

**Figure 5 ijms-23-09567-f005:**
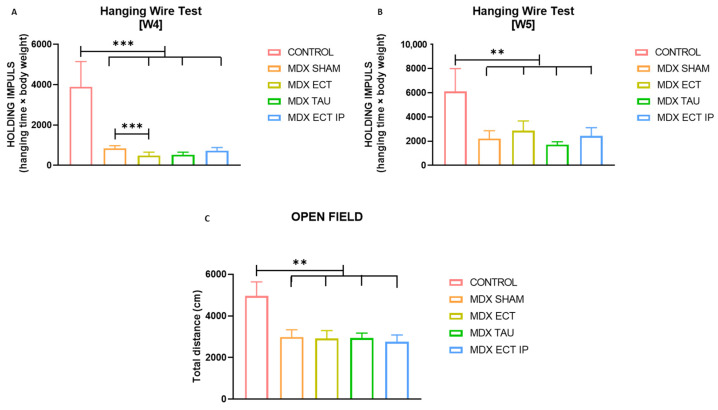
Results of functional tests in mice (**A**). Holding impulse (hanging time × body weight) of the four limb hanging wire in 4-week-old mice for evaluation of muscle strength. (**B**) Holding impulse of the four limb hanging wire in 5-week-old mice. (**C**) Total distance travelled in the open field test in 5-week-old mice for evaluation of locomotion. Data are presented as mean values ± SEM. Data of the hanging wire test were log-transformed to adhere to normality assumptions. A more detailed overview of statistical analysis is provided in Appendix A. Statistical analysis was carried out to detect significant differences between sham-treated *mdx* mice and other groups or between control mice and other groups. Significant differences are annotated by ** (*p* < 0.01) and *** (*p* < 0.001).

**Figure 6 ijms-23-09567-f006:**
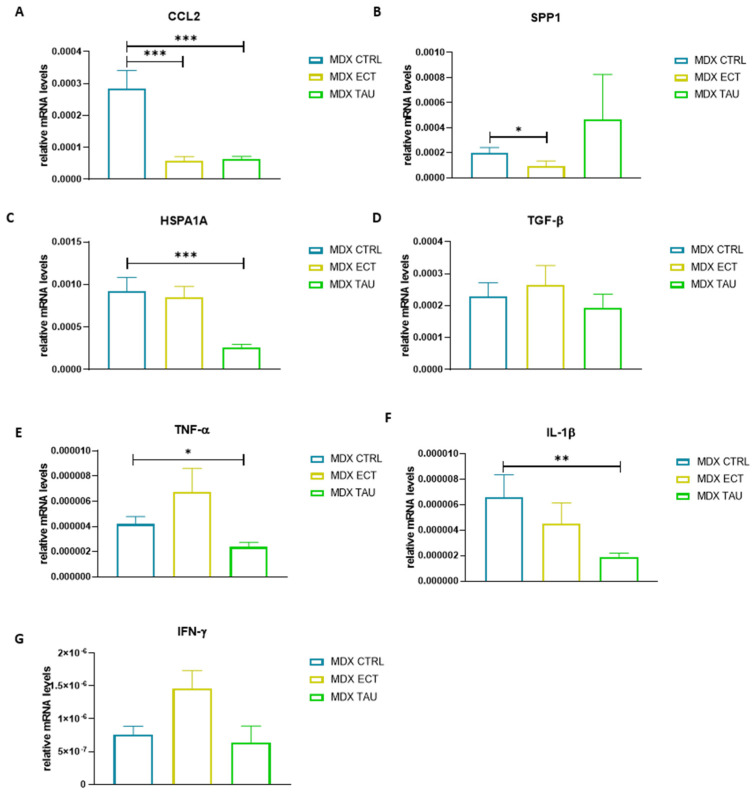
Relative gene expression of inflammatory markers in treated and untreated *mdx* mice. Graphical representation of mRNA levels of *CCL2* (**A**), *SPP1* (**B**), *HSPA1A* (**C**), *TGF-β* (**D**), *TNF-α* (**E**), *IL-1β* (**F**) and *IFN-γ* (**G**). Relative gene expression is normalized to the housekeeping gene *GAPDH* and expressed as 2^−ΔCT^, and data are presented as 2^−ΔCT^ mean ± standard error of the mean (SEM). Data underwent a log-transformation in order to adhere to normality assumptions. A more detailed overview of statistical analysis is provided in Appendix A. Statistical analysis was carried out to detect differences between untreated *mdx* mice and other groups. Significant differences are annotated by * (*p* < 0.05), ** (*p* < 0.01) and *** (*p* < 0.001).

**Figure 7 ijms-23-09567-f007:**
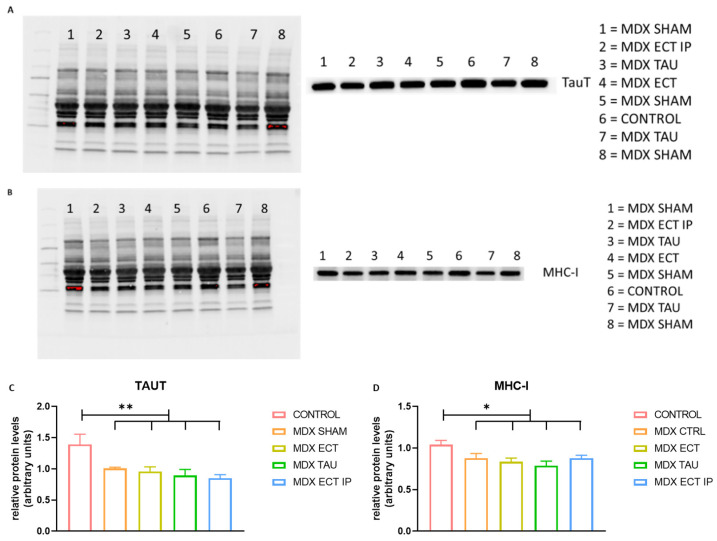
Protein expression of TauT and MHC-I in control and *mdx* mice. (**A**) Representative images of the Stain-Free blot (left) used for total protein normalization and TauT protein bands (right) as imaged through chemiluminescent detection. (**B**) Representative images of the Stain-Free blot (left) used for total protein normalization and MHC-I protein bands (right) as imaged through chemiluminescent detection. (**C**) Graphical representation of TauT protein levels in control and treated *mdx* mice after normalization against total protein using Bio-Rad stain-free gels. (**D**) MHC-I protein levels in control and treated *mdx* mice after normalization against total protein. Data are presented as mean ± SEM. Statistical analysis evaluated differences between sham-treated *mdx* mice and other groups or between control mice and other groups. Significant differences are annotated by * (*p* < 0.05); ** (*p* < 0.01).

**Table 1 ijms-23-09567-t001:** Mouse phenotype data. Body weight, body length and serum CK values were determined. Mean values ± SEM are reported for both sexes in each group to allow better interpretation.

	N			Body Weight (g) ± SEM		Body Length (cm) ± SEM		CK (U/L) ± SEM	
	♂	♀	Total	♂	♀	♂	♀	♂	♀
CTRL	4	3	7	22.3 ± 0.5	20.0 ± 0.0	9.6 ± 0.1	9.2 ± 0.2	2633 ± 523	1668 ± 364
MDX SHAM	4	4	8	22.8 ± 0.9	19.8 ± 0.6	9.3 ± 0.1	8.8 ± 0.1	16,000 ± 3230	12,136 ± 1765
MDX ECT	4	7	11	20.5 ± 0.6	17.7 ± 0.4	9.2 ± 0.1	8.7 ± 0.1	15,700 ± 5194	9263 ± 890
MDX TAU	7	4	11	23.3 ± 0.6	18.3 ± 0.5	9.3 ± 0.1	8.8 ± 0.1	14,313 ± 1690	9976 ± 2199
MDX ECT IP	5	4	9	21.6 ± 0.6	17.3 ± 0.9	8.8 ± 0.1	8.6 ± 0.2	19,035 ± 4806	9862 ± 2254

## Data Availability

Data presented in this study are available from the corresponding author on reasonable request.

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
