# Peer review of "Exploring the Therapeutic Potential of Ectoine in Duchenne Muscular Dystrophy: Comparison with Taurine, a Supplement with Known Beneficial Effects in the mdx Mouse"

_ijms, 2022, doi:10.3390/ijms23179567_

Round 1

Reviewer 1 Report

Overall Comments: The manuscript by Merckx et al., centers on the testing of the amino acid product ectoine in the Duchenne muscular dystrophy (DMD) mdx mouse model. Ectoine is a natural product amino acid produced from bacteria and thought to protect skin cells from UV damage among other cytoprotective properties. The authors compare testing of this ectoine compound with taurine supplementation, which previously has been shown to benefit mdx mice. The authors tested both in vitro (CCL-136 cells) and in vivo (mdx mice) if ectoine could be used as an alternative to taurine supplementation to improve overall mdx muscle histology and outcomes. Taurine and ectoine acts as an osmolyte to regulate fluids in the cells that comprise the muscle. Both acted to improve molecular markers, as defined via quantitative PCR (qPCR) markers in the mdx mice. The taurine and ectoine treated-mdx mice improved histological markers but not functional parameters. The authors conclude that ectoine is a capable substitute for taurine with regards to improvement of mdx muscles.

The manuscript objectives are overall reasonable, with perhaps the exception of testing the rhabdomyosarcoma cells. The authors should provide more justification, in addition to an actual citation in the manuscript, for the use of this cell line over a DMD muscle cell line. Additional clarifications over key aspects of the methodology are warranted to put these studies in proper perspective for the field.

Major Comments:

1. It appears that all qPCR data was performed as an absolute measurement. Is this correct? Why wasn’t a normalization housekeeping (e.g. beta-actin) control not used? The expression values listed (e.g. Figure 6) seem to be very minor and it is not clear if some factors (6F; IL-1beta) are significant given how low in expression the transcript appears to be in abundance.

2. My biggest question is why was a rhabdomyosarcoma line (e.g. CCL-136) used for the taurine and ectoine-testing and not a DMD muscle cell line? Maybe a muscle line from DMD patients and/or mdx mice?

3. Figure 1 is not directly referred to in the text? Supplemental figure 1 is as is some of the data from figure 1. Please correct.

4. Figure 4 (A-D), the H&E images are a little too bright and difficult to interpret. Can the authors tone down the intensity and please add a scale bar for appropriate reference?

5. More methodological justification and/or description on the identification of necrotic versus regenerating muscle fibers (Figure 4E) need to be clarified. Please state the quantification method(s) and cite a source for this approach.

6. Minor comment. Please clarify the day of the initiation of the drug trials. Is day 7, postnatal day 7? Have the mice not yet been weaned from their mothers? There needs to be some experimental clarifications in the Materials and Methods section (4).

7. Lastly, although it is not explicitly stated in the Methods, can the authors state the sex of the mice used in the experiments, in particular the drug experiments?

Reviewer 2 Report

This manuscript indicated effect of ectoine in mdx mice as alternative of glucocorticoid. This report is very interesting for suppretion of inflammation in mdx mice. But some corrections may be required. In introduction,  it is better to add molucular mechanusms of glucocorticoid in DMD already reported in previous manuscripts.  In Figure 1, it is better to analyze the apoptosis such as caspase, annexinV,  TUNEL, and the cell cycle for each phases, G1, G2, M, and S.  In figure 2, it is better to analyze the expression of inflammatory markers by protein level. In table 1, it is better to analyze microRNAs (miR-1, miR-133a, miR-206) as another biomarkers for serum CK.In figure 4, It is better to examine  number of central nuclei fibers, and size distribution of myofiber. In figure 5, it is better to analyze the functional tests using more older mice. In fugure 6, it is better to analyze the wild type control mice.

Round 2

Reviewer 1 Report

Since the original submission the authors have made some improvements with regards to the clarity of the methodology. I appreciate the adjustments to the images in figure 4 for visual clarity.  I appreciate the in text methodology descriptions and better clarity with regards to quantification of regenerating fibers and qPCR controls used. I have no additional edits.

Reviewer 2 Report

This manuscript was corrected according to reviewer's comments